# CEMIP, a Promising Biomarker That Promotes the Progression and Metastasis of Colorectal and Other Types of Cancer

**DOI:** 10.3390/cancers14205093

**Published:** 2022-10-18

**Authors:** Kevin Domanegg, Jonathan P. Sleeman, Anja Schmaus

**Affiliations:** 1European Center for Angioscience (ECAS), Medical Faculty Mannheim, University of Heidelberg, 68167 Mannheim, Germany; 2Institute of Biological and Chemical Systems-Biological Information Processing, Karlsruhe Institute of Technology (KIT) Campus Nord, 76344 Eggenstein-Leopoldshafen, Germany

**Keywords:** CEMIP, KIAA1199, hyaluronic acid, hyaluronidase, metastasis, EMT

## Abstract

**Simple Summary:**

CEMIP (cell migration-inducing and hyaluronan-binding protein) has been implicated in the pathogenesis of numerous diseases, including colorectal and other forms of cancer. The molecular functions of CEMIP are currently under investigation and include the degradation of the extracellular matrix component hyaluronic acid (HA), as well as the regulation of a number of signaling pathways. In this review, we survey our current understanding of how CEMIP contributes to tumor growth and metastasis, focusing particularly on colorectal cancer, for which it serves as a promising biomarker.

**Abstract:**

Originally discovered as a hypothetical protein with unknown function, CEMIP (cell migration-inducing and hyaluronan-binding protein) has been implicated in the pathogenesis of numerous diseases, including deafness, arthritis, atherosclerosis, idiopathic pulmonary fibrosis, and cancer. Although a comprehensive definition of its molecular functions is still in progress, major functions ascribed to CEMIP include the depolymerization of the extracellular matrix component hyaluronic acid (HA) and the regulation of a number of signaling pathways. CEMIP is a promising biomarker for colorectal cancer. Its expression is associated with poor prognosis for patients suffering from colorectal and other types of cancer and functionally contributes to tumor progression and metastasis. Here, we review our current understanding of how CEMIP is able to foster the process of tumor growth and metastasis, focusing particularly on colorectal cancer. Studies in cancer cells suggest that CEMIP exerts its pro-tumorigenic and pro-metastatic activities through stimulating migration and invasion, suppressing cell death and promoting survival, degrading HA, regulating pro-metastatic signaling pathways, inducing the epithelial–mesenchymal transition (EMT) program, and contributing to the metabolic reprogramming and pre-metastatic conditioning of future metastatic microenvironments. There is also increasing evidence indicating that CEMIP may be expressed in cells within the tumor microenvironment that promote tumorigenesis and metastasis formation, although this remains in an early stage of investigation. CEMIP expression and activity can be therapeutically targeted at a number of levels, and preliminary findings in animal models show encouraging results in terms of reduced tumor growth and metastasis, as well as combating therapy resistance. Taken together, CEMIP represents an exciting new player in the progression of colorectal and other types of cancer that holds promise as a therapeutic target and biomarker.

## 1. Introduction

CEMIP (cell migration-inducing and hyaluronan-binding protein) was first identified in 1999 in a cDNA screen for large then-unknown proteins greater than 50 kDa in size and was originally called KIAA1199 [1]. In different contexts, CEMIP has also been called colon cancer secreted protein 1 (CCSP1), transmembrane protein 2-like protein (TMEM2L), hyaluronan binding protein involved in hyaluronan depolymerization (HYBID), and hypothetical protein IR2155535. Four years after its discovery, CEMIP came to the fore as a gene expressed in Dieters’ cells and fibrocytes in the inner ear, mutations in which are associated with hereditary hearing loss [2]. A decade later, the discovery that CEMIP possesses a hyaluronidase activity gave a first indication of its functional significance [3]. Since then, a number of additional molecular functions have been ascribed to the CEMIP protein, and it has been implicated in the pathogenesis of a variety of diseases. Most significantly, it is increasingly recognized that CEMIP plays an important role in the genesis, progression, and metastasis of a growing number of cancer types, among which is colorectal cancer [4]. 

Compared with normal tissue, CEMIP expression is strongly upregulated in many types of cancer. For example, CEMIP expression in breast cancer is between 2- and 8-fold higher than in normal breast tissue, depending on the molecular subtype [5]. In the context of colorectal cancer (CRC), it is notable that CEMIP has been identified in several screens as one of the most highly upregulated genes in tumors compared with normal tissues [6,7,8,9,10]. Mean transcript levels for CEMIP are more than 30-fold higher in both microsatellite stable and instable colorectal adenocarcinomas than in normal colon mucosa [7]. In GEO dataset analyses using various publicly available datasets, CEMIP has been identified as one of the top 10 most significantly upregulated genes in CRC compared with normal colorectal tissue [11]. In microarray analyses, CEMIP has been indicated as the second most strongly overexpressed gene [12]. Functional genomic mRNA profiling has also identified CEMIP as one of the top overexpressed genes [13]. 

Consistent with its strongly increased expression in colorectal tumors, CEMIP has been proposed as a potential blood-borne biomarker for CRC. High CEMIP levels are secreted by benign colorectal adenomas and remain high in different CRC tumor stages, suggesting that CEMIP has potential as an early diagnostic marker [7,9,11]. Moreover, CEMIP RNA levels have been shown to be significantly higher in the blood from patients with colorectal adenomas or more advanced CRC than in healthy controls [9], indicating that CEMIP could serve as a plasma RNA biomarker in this context. High levels of CEMIP are also secreted from other types of cancer. Accordingly, CEMIP has been suggested as a diagnostic serum marker for cholangiocarcinoma and pancreatic cancer [14,15]. 

CEMIP is more than just a highly upregulated biomarker in CRC, as the enhanced expression of CEMIP has also been associated with poor prognosis in colorectal cancer patients [8,11,12,16] and is an independent prognostic factor in this context [11,16]. Some studies have indicated that CEMIP expression is further increased in metastatic lesions of CRC compared with the primary tumor [8,17,18]. A broad body of evidence also documents the association between CEMIP expression, and poor prognosis and metastasis in other types of human cancer, including lung [19,20,21], gastric [22,23,24,25], pancreatic [26], brain [27,28], breast [5,10,29,30], and papillary thyroid [31] cancers, as well as in cholangiocarcinoma [14], hepatocellular carcinoma (HCC) [32,33], laryngeal [34], and oral squamous cell carcinoma [35]. Again, CEMIP has also been shown to be an independent prognostic factor in NSCLC [19,21], gastric cancer [24,25], pancreatic ductal adenocarcinoma [26], laryngeal squamous cell carcinoma [34], osteosarcoma [36], and HCC [33]. 

The correlation between CEMIP expression and poor prognosis for CRC cancer patients reflects the fact that CEMIP plays a functional role in the process of metastasis, as evidenced by loss- and gain-of-function studies using experimental and spontaneous colorectal metastasis models. The knockdown of CEMIP has been shown to significantly reduce metastasis to the lung, liver, and various other organs [8,18,37]. Conversely, the increased expression of CEMIP stimulates liver metastasis, and acts cooperatively with glutaminase 1 [18,38]. Similar observations supporting a functional role for CEMIP in metastasis have been made using hepatocellular, gastric, and breast tumor models in vivo [23,29,39,40,41]. For example, CEMIP expression has been reported to stimulate the metastasis of sorafenib-resistant HCC cells [41]. In different breast and gastric tumor models, CEMIP expression has been found to promote metastasis to the lung and liver, although this might be due at least in part to concurrent accelerated primary tumor growth [23,29,39]. The CRISPR-Cas9-mediated ablation of CEMIP in brain-tropic breast cancer cells has been shown to inhibit the formation of brain metastases following intra-cardiac injection [27].

The above observations suffice to illustrate the functional association between CEMIP expression, and tumorigenesis and metastasis formation. Many of the data documenting this association stem from investigations in the context of CRC but are also observed for many other types of cancer. These data suggest that by understanding the molecular mechanisms through which CEMIP promotes tumor progression and metastasis, new avenues for therapeutic intervention might be revealed. Although numerous mechanisms have been suggested, the way in which CEMIP contributes at the molecular level to tumor progression and metastasis is still poorly understood. Studies to date have focused almost exclusively on the tumor cell-specific functions of CEMIP, yet given the biology of CEMIP in other physiological and pathophysiological contexts, the expression of CEMIP in cells of the tumor microenvironment such as fibroblasts may be equally as important for the tumor- and metastasis-promoting role of CEMIP. However, the relevance of CEMIP expression in the tumor microenvironment remains largely to be investigated. Here, we review our current understanding of the mechanisms through which CEMIP contributes to tumor progression and suggest ways in which the perturbation of CEMIP expression and function might be leveraged therapeutically.

## 2. Regulation, Molecular Architecture, and Cellular Distribution of CEMIP

The human *CEMIP* gene is located on chromosome locus 15q25.1 (at 80,779,370–80,951,771 [42]), where it spans approximately 170 kb, and consists of 29 exons [2,43]. The *CEMIP* promoter includes several predicted *cis*-acting elements for transcriptional regulators, such as activator protein-1 (AP-1), Twist-1, GLI, ATF3, and nuclear factor-κB (NF-κB) [44,45,46]. The CEMIP locus is further surrounded by four T-cell factor 4 (TCF4)–binding sites, reflecting the fact that CEMIP is a target of the Wnt signaling pathway [6,47,48]. It also contains a hypoxia response element, consistent with the increased CEMIP expression observed under low-oxygen conditions through the stabilization of hypoxia-inducible-factor-2α (HIF-2α) [17,49]. The basic promoter activity of CEMIP is also regulated by DNA methylation and in human breast cancer hypomethylation correlates with high CEMIP expression [10,44]. Interestingly, low oxygen levels can also inhibit the function of the histone demethylase Jarid1A, which normally removes methyl groups from H3K4me3 within the *CEMIP* promoter, resulting in upregulated CEMIP transcription [17]. Additional epigenetic regulation is mediated by a number of microRNAs that, for example, bind directly to the 3′-UTR of the CEMIP mRNA and thus regulate its expression, including miR-486-5p [20,31], miR-486-3p [50], miR-216a [8], miR-140-5p [51], miR-29c-3p [23], miR-4656 [52], miR-148-3p [53], miR-4677-3p [54], miR-17-5p [45], and miR-4306 [55]. A further layer of complexity in the regulation of CEMIP expression is provided by lncRNAs such as LINC00958, CASC19, and HCP5, which have been shown to neutralize the effect of some of these miRNAs, leading to increased CEMIP expression [51,52,55].

The *CEMIP* gene encodes a 1361 aa protein with a 30 aa cleavable signal peptide at the N-terminus, resulting in a mature 1331 aa protein with a calculated weight of 153 kDa [2]. Various domains have been identified in the protein (Figure 1). The N-terminal signal sequence is essential for the N-glycosylation of CEMIP, the proper translocation of CEMIP as well as its HA depolymerizing activity [56]. CEMIP contains two GG domains, one G8 domain, and four PbH1 domains [57,58,59]. The G8 domain (44 aa-166 aa), named after its eight conserved glycine residues, contains five repeated β-strand pairs and has been hypothesized to be involved in extracellular ligand binding [58]. Recently, the CEMIP G8 domain has been found to bind to annexin A1 (ANXA1), enabling CEMIP to adhere to the cell membrane of ANXA1-positive cells [60]. The two GG domains are composed of seven β-strands and two α helices and are located N-terminally after the G8 domain and at the C-terminus [57]. Currently, the GG domain is of unknown function but has been suggested to be possibly involved in the HA degrading activity of CEMIP [3]. The four PbH1 domains each span 22-23 aa, are predicted to function in polysaccharide hydrolysis, and are located centrally in the protein [61,62]. Furthermore, a motif termed the B domain, spanning from Ala295 to Thr591, is necessary for endoplasmic reticulum (ER) retention, presumably compensating for the lack of an ER retention signal (KDEL) [29]. 

The multi-domain protein CEMIP has emerged as a promiscuous interaction partner for an increasing number of proteins (Table 1). The domains of the CEMIP protein responsible for these interactions have not always been investigated, but interestingly, binding partners have been described for all regions of the protein. This suggests that CEMIP acts as scaffold that is involved in the recruitment and organization of a range of other proteins. However, the direct functional molecular consequences of these interactions are incompletely understood. Some examples are given in the text below.

At the cellular level, the CEMIP protein can be distributed throughout the cytoplasm [63] and localized to the ER [29,59] and is found in different endosomes [47]. The protein can also be associated with the cell membrane, is a constituent of exosomes, and is abundantly secreted [12,14,27,29,56,59]. The nuclear localization of the CEMIP protein has also been suggested in colon cancer cell lines [59] and has also been reported in samples from a subset of colorectal cancer patients, where nuclear localization has been associated with a reduced incidence of lung and liver metastases [7].

**Table 1 cancers-14-05093-t001:** Summary of the molecular interactions of CEMIP.

Protein	Interaction with CEMIP Domain or Region	Evidence for Complex Formation with CEMIP	Proposed Function of CEMIP	Reference
Epidermal growth factor receptor (EGFR)	N-terminus (60 aa, including part of G8 domain)	Co-IP with full-length CEMIP, N- and C-terminal mutants	EGFR stability and signaling	[64]
Plexin A2	N-terminus (100 aa, including part of G8 domain)	Co-IP with full-length CEMIP, N- and C-terminal mutants; PLA	Protection from Semaphorin 3A-Plexin A2-dependent cell death	[64]
Annexin A1 (ANXA1)	G8	IP and MS; co-IP (also using G8-deleted CEMIP mutants)	Adherence to cell membranes and HA degradation	[60]
Binding immunoglobulin protein (BiP)	B domain (aa 295–591)	IP and MS; co-IP	ER retention and cell migration	[29]
O-GlcNAc transferase (OGT)	PbH1 domains (572–819 aa)	Co-IP; PLA	Elevated O-GlcNAcylation of β-catenin, glutamine metabolic reprogramming	[38]
Beta-catenin (β-catenin)	820–1204 aa	Co-IP; PLA	Enhanced β-catenin nuclear translocation (via OGT-mediated O-GlcNAcylation), glutamine metabolic reprogramming	[38]
Coatomer protein complex α-subunit (COPA)	Second GG domain (1201–1361 aa)	MBP-tag pull down assay, MS, co-IP	Not addressed in the study (transport of CEMIP to ER?)	[65]
Glycogen phosphorylase kinase β-subunit (PHKB)	Second GG domain (1201–1361 aa)	MBP-tag pull down assay, MS, co-IP	Glycogen breakdown and cancer cell survival	[65]
PP2A	C-terminus (880–1362 aa)	IP and MS; co-IP	Enhancing phosphatase activity of PP2A leading to dephosphorylation of stathmin, microtubule destabilization, and enhanced cell motility	[37]
Clathrin heavy chain (CHC)	Not addressed	Co-IP	Clathrin-mediated endocytosis and HA degradation	[3]
Ephrin A2 (EPHA2)	Not addressed	IP and MS; co-IP	Not addressed in the study (migration/repulsion of cells?)	[59]
Inositol 1,4,5-triphosphate receptor 3 (ITPR3)	Not addressed	IP and MS; co-IP	Calcium ion transport into cytosol, signaling pathway activation, ferroptosis protection	[59,66]
Mitogen-activated protein kinase kinase 1 (MEK1)	Not addressed	Co-IP	Sustained MEK1-ERK1/2 activation	[47]
Protein tyrosine phosphatase 4A3 (PTP4A3)	Not addressed	IP and MS; co-IP	Activation of EGFR signaling	[23]
TGFBR1 and TGFBR2	Not addressed	Co-IP	Promotion of TGFβ signal transduction	[18]
WW domain binding protein 11 (WBP11)	Not addressed	IP and MS; co-IP	Activation of FGFR expression and Wnt/β-catenin signaling	[23]

IP: immunoprecipitation. Co-IP: co-immunoprecipitation. MS: mass spectrometry. PLA: proximity ligation assay.

In the healthy organism, CEMIP is found in many different tissues and is strongly expressed in the brain (especially the hippocampus), placenta, lung, and testis but is virtually absent in tissues such as breast or liver [43,65,67,68]. Fibroblasts are thought to be among the main producers of CEMIP, although a number of other cell types, such as immune and stem cells, chondrocytes, astrocytes, endothelial cells, and alveolar type II cells, also express the protein [10,46,68,69]. Compared with normal tissue, CEMIP expression is strongly upregulated in cancer tissues from various organs, most notably in CRC (see above) [10,11].

## 3. CEMIP Functions and Their Relevance for Tumor Growth and Metastasis

In colon cancer cells, the expression of CEMIP influences the expression of genes linked to cell-cycle regulation, proliferation, migration, and apoptosis [7]. Consistently, CEMIP has been implicated in various fundamental aspects of tumor biology. For example, the expression of CEMIP has been implicated in promoting tumor cell proliferation and survival. CEMIP knockdown in breast cancer cells results in reduced proliferation in vitro, and when CEMIP knockdown cells are injected into the mammary fat pads of female athymic nude mice, there is a significant decrease in tumor incidence and growth [39]. The knockdown of CEMIP also reduces the ability of human colon cancer cells to form xenograft tumors in athymic mice [12]. Consistently, the overexpression of CEMIP in gastric cancer cells promotes tumor growth in vivo upon subcutaneous or intra-peritoneal injection [23]. Mechanistic studies in hepatocellular and colorectal cancer cells have shown that CEMIP knockdown leads to decreased levels of cyclin D1 and E1, hindering cell-cycle progression, increasing apoptosis, and attenuating tumor growth and metastasis [40,70]. Furthermore, the loss of CEMIP represses the major ER chaperone GRP78/BiP, which is associated with attenuated unfolded protein response (UPR) and increased levels of ER stress markers [40,70]. As the UPR can ultimately counteract ER-stress-induced apoptosis, these observations may in part explain the survival function of CEMIP [71].

With regard to the process of metastasis, CEMIP has been implicated in regulating cell migration and invasion. For example, the knockdown of CEMIP in colorectal and gastric cancer cells inhibits their migration and invasiveness in vitro [8,72,73]. CEMIP-enhanced cell migration requires endoplasmic reticulum (ER) localization, where it forms a stable complex with the chaperone binding immunoglobulin protein BiP [29]. Hypoxia stimulates CEMIP-dependent motility in colorectal and PDAC cancer cells [17,49]. Furthermore, CEMIP induces invasion in NSCLC and ovarian cancer cells in a manner that is associated with the increased expression of PI3K and AKT [19,74]. Moreover, the physical association between CEMIP and the protein phosphatase 2A (PP2A) increases the phosphatase activity of PP2A in colorectal cancer cells, resulting in decreased phosphorylation of the microtubule-destabilizing protein stathmin, which enhances cell motility [37]. 

In addition to these rather general observations, CEMIP has also been ascribed specific functions, many of which could potentially contribute to its tumor- and metastasis-promoting functions, as outlined below.

### 3.1. Hyaluronan Depolymerization

Hyaluronan (HA) is a large, linear polymer composed of repeating disaccharide units of glucuronic acid and N-acetylglucosamine. It is the major constituent of the extracellular matrix (ECM) [75]. Despite its simple structure, HA has a variety of biological functions, including the regulation of morphogenesis, proliferation, and migration [76,77]. It serves to structure the interstitial matrix through interacting with HA-binding proteins and acts as a sink for sodium, thereby contributing to the regulation of osmolarity and blood pressure [78,79]. HA can also be covalently modified by the heavy chains of the serum protein inter-alpha-inhibitor (IαI) through the activity of tumor necrosis factor-inducible gene 6 protein (TSG-6), which plays an important role in regulating inflammation [80]. 

The biological activity of HA is dependent on its concentration and molecular size. Under normal physiological conditions, HA is mainly present in tissues in a high-molecular-weight form (HMW-HA) that is continuously degraded and turned over. The activity of the HA synthases (HAS) and hyaluronidases (HYAL) responsible for the synthesis and turnover of HA is tightly coordinated to ensure that sufficient HA is produced but without the accumulation of HA degradation products. Under certain pathological conditions, such as tissue injury and cancer, this coordination can be perturbed, resulting in the accumulation of HA oligosaccharides of 4–25 disaccharides in length. These HA oligosaccharides act as damage-associated molecular patterns (DAMPs) by exerting a number of inflammation-associated biological activities not observed with HMW-HA. In the context of cancer, this accumulation is often associated with the increased synthesis of HA in conjunction with the enhanced hyaluronidase-mediated cleavage of the HA and the activity of free radicals (reviewed in [77,81,82]). Receptors such as TLR-4, CD44, and LYVE-1 mediate the DAMP activities of HA oligosaccharides, which include the increased expression of matrix metalloproteases and pro-inflammatory cytokines, the activation of dendritic cells [83,84], the polarization of macrophages [85], the induction of inflammatory responses [86], and the stimulation of angiogenesis and lymphangiogenesis [87,88]. 

CEMIP has been demonstrated to foster the depolymerization of HMW-HA into low- and intermediate-sized HA oligosaccharides (LMW/IMW-HA) [3,89,90] and the enhanced expression of CEMIP in pancreatic tumor cells leads to increased levels of HA oligosaccharides, which is associated with increased invasive behavior [91]. Conversely, the loss of CEMIP increases the levels of HMW-HA, both in glioblastoma cells in vitro and in the respective tumors in vivo, which is linked to reduced tumor growth [28]. However, the mechanism behind the CEMIP hyaluronidase activity remains only partially understood. The structure of CEMIP shows no significant homology to known hyaluronidases, and also lacks the HA-link modules and B(X7)B HA-binding motifs that are found in other HA binding proteins, such as CD44, RHAMM, and link proteins [92,93,94]. Although the secretion of CEMIP from cells is necessary for hyaluronidase activity [56], no hyaluronidase activity is detected in cell-free CEMIP expression systems or in CEMIP-rich conditioned media [3,56], and the presence of cells is required for hyaluronidase activity to be observed. It is thought that secreted CEMIP binds to HMW-HA in the extracellular space, and that internalization via clathrin-mediated endocytosis and endolysosomal degradation may then ensue. As degradation fragments are found extracellularly, presumably, they are subsequently expelled into the extracellular milieu. On the other hand, the tethering of CEMIP to the cell membrane via ANXA1 has recently been found to be necessary and sufficient for CEMIP-mediated HA degradation [60]. A proposed mechanism for the HA-degrading activity of CEMIP is shown in Figure 2. 

The hyaluronidase activity of CEMIP may account for its tumor- and metastasis-promoting role in a number of ways. CEMIP has been reported to be localized at the invasive front of colon cancer tissue [17], and the hyaluronidase activity may conceivably contribute to tissue invasion through the degradation of HA-dependent migratory barriers. Interestingly, ANXA1 has been reported to positively regulate cellular motility in pancreatic cancer [95], which could conceivably link the CEMIP hyaluronidase activity with increased motility, although this remains to be investigated. Perhaps most significantly, LMW/IMW-HA produced by CEMIP is likely to have several metastasis-promoting functions, including the stimulation of migration and invasion [84,91,96,97], and has been implicated in CEMIP-induced migration [91]. These oligosaccharides can also induce the expression of a number of different matrix metalloproteases (MMPs) in both cancer cells and in cells associated with the tumor microenvironment, such as fibroblasts, macrophages, and endothelial cells [81,87,98,99,100]. MMPs can degrade components of the extracellular matrix to facilitate tumor invasion into the surrounding tissue, and can also release signaling molecules from the ECM to induce the epithelial–mesenchymal transition (EMT) [101]. The growth factors and cytokines produced in response to HA oligosaccharides can also induce a metastasis-promoting inflammatory environment (see below), which is associated with activation of the innate immune system [102] and the M2 polarization of macrophages [85]. HA oligosaccharides also induce angiogenesis (reviewed in [81]), which promotes tumor growth and metastasis formation in a number of ways [100]. Furthermore, the HA-oligosaccharide-mediated induction of lymphangiogenesis via LYVE-1 may contribute to tumor-induced lymphangiogenesis, leading to increased dissemination to regional lymph nodes [88], consistent with the observation that HA oligosaccharide levels in the interstitial fluid of colorectal cancers correlate with the incidence of lymph node metastasis formation [103].

### 3.2. Signaling

CEMIP has been reported to play a crucial role in the regulation of several signaling pathways, mainly in the context of tumor cells (Figure 3). Examples include Wnt/β-catenin [6,7,11,22,23,45,47,59,73,104,105,106,107], EGFR [20,23,41,64], and downstream pathways such as PI3K-Akt [19,74,108], MEK/ERK [47,109,110], STAT3 [30,41], and PP2A/stathmin [37]. Although not always explicitly investigated, the roles that these pathways are known to play in the process of tumorigenesis and metastasis suggest that CEMIP may contribute to the growth, dissemination, and spread of cancer through regulating these pathways. 

#### 3.2.1. Reciprocal Regulation between Wnt/β-Catenin Signaling and CEMIP Expression

The deregulation of the Wnt pathway is a hallmark of CRC progression, and Wnt signaling drives the initiation and progression of CRC. Furthermore, Wnt signaling has been directly implicated in stimulating metastasis formation, for example, through the induction of EMT, anchorage-independent growth, and therapy resistance [111]. Several observations indicate that CEMIP positively regulates Wnt signaling. In transcriptomic profiles from normal colonic mucosa and colorectal adenomas, there is a strong positive correlation between CEMIP expression and the Wnt/β-catenin signaling pathway, which is recapitulated in colon cancer cell lines with loss and gain of function for CEMIP [6,7,11,38]. At the functional level, the depletion of CEMIP in CRC cells is able to block Wnt signaling, the expression of EMT-related genes, proliferation, migration, and invasion [7,11,73,112]. Consistently, the overexpression or silencing of CEMIP in other cell types, such as gastric cancer cell lines, vascular smooth muscle cells, osteoblastic stem cells, and chondrocytes, also activates or attenuates Wnt/β-catenin signaling, respectively, presumably through the CEMIP-regulated expression and stability of β-catenin [22,23,104,105,107]. Wnt signals activate the Frizzled receptors, thereby inducing the formation of a DSH-β-catenin complex, and it is thought that the formation of this complex as well as the subsequent translocation of β-catenin into the nucleus is enhanced in the presence of CEMIP through several mechanisms. First, CEMIP suppresses the expression of axin2, a negative regulator of β-catenin [23]. In addition, CEMIP has been shown to foster the O-GlcNAcylation of β-catenin by acting as an adaptor protein for O-GlcNAc transferase (OGT). O-GlcNAcylation stabilizes β-catenin [113], which results in increased nuclear translocation and transcriptional activity of β-catenin [38]. Furthermore, CEMIP has been reported to form a complex with WW domain binding protein 11 (WBP11), which increases the expression of Wnt/β-catenin signaling target genes [23], although the mechanism underlying this observation remains obscure. 

Interestingly, CEMIP expression itself is positively regulated by Wnt signaling, suggesting that a positive feedback loop exists between Wnt signaling and CEMIP expression. Notably, TCF4 acts as transcription factor in Wnt/β-catenin signaling, and the CEMIP locus is surrounded by four TCF4-binding sites [6,48,114]. Consistently, the expression of CEMIP in colorectal cancer cells is reduced by the dominant-negative form of TCF4 and increased by β-catenin-dependent transcription [6,11,38,48]. In BRAFV600E-mutated colorectal cancers, CEMIP expression is increased in a β-catenin-dependent manner, which contributes to acquired resistance to Mitogen-activated protein kinase kinase 1 (MEK1) inhibition [47]. Moreover, the β-catenin-dependent upregulation of CEMIP has been reported in anoikis-resistant prostate cancer cells, although in this case via crosstalk with AMPK/GSK3β [115]. Importantly, the reciprocal regulation of CEMIP and Wnt/β-catenin signaling enhances the glutamine metabolic reprogramming of colorectal cancer cells, which fosters metastasis [38]. 

#### 3.2.2. Regulation of EGFR Signaling by CEMIP

EGFR is a member of the ErbB family of tyrosine kinase receptors, which signals via ligands such as epidermal growth factor (EGF), transforming growth factor-alpha (TGF-α), amphiregulin, heparin-binding EGF (HB-EGF), and betacellulin. As a consequence, a number of downstream signaling cascades are activated, such as RAS/RAF/MEK/MAPK/ERK, phosphatidylinositol 3-kinase (PI3K) and Akt, protein kinase C (PKC), Src, and the JAK/STAT pathways. These pathways foster metastasis, as well as a number of other tumorigenic properties [116]. For example, EGFR signaling is able to initiate EMT and stimulate the formation of lamellipodia, filopodia, and invadopodia, which facilitate the migration and invasion of cancer cells, as well as promote the survival and outgrowth of disseminated tumor cells at secondary sites [117]. EGFR signaling is also involved in the progression of CRC, and the anti-EGFR monoclonal antibodies cetuximab and panitumumab are approved for the targeted therapy of metastatic CRC [118].

CEMIP impacts EGFR activity in a number of ways, including increasing the total cellular levels of EGFR [119] and promoting EGFR signaling. In this latter regard, CEMIP directly binds to EGFR, which enhances the phosphorylation of EGFR and protects it from lysosomal degradation, thereby promoting EGFR signaling [64]. Furthermore, CEMIP also activates the EGFR pathway by directly binding to protein tyrosine phosphatase 4A3 (PTP4A3) [23], a phosphatase that hyper-activates EGFR [23,120]. Consistent with these observations, CEMIP loss of function impairs the EGF-dependent phosphorylation of Src, MEK1, ERK1/2, Akt, and Stat3 [20,41,64]. CEMIP also protects cells from semaphorin 3A-mediated cell death by binding to its transmembrane receptor Plexin A2 (PLXNA2) and by stimulating EGFR signaling [64]. 

#### 3.2.3. Regulation of BiP Expression by CEMIP and Its Role in Ca^2+^ Signaling

Chaperone binding immunoglobulin protein (BiP), also known as GRP78, is a member of the heat shock protein 70 (HSP70) family whose role is to maintain correct intracellular protein folding. CEMIP has been reported to increase BiP expression through as-yet-unknown mechanisms [70,121]. In CRC cells, the experimental knockdown of CEMIP inhibits the growth of cells and induces apoptosis, which is linked to reduced levels of BiP and the attenuation of the UPR [70]. CEMIP-induced BiP expression has also been found to increase cell survival under hypoxic conditions [121]. As CEMIP expression is upregulated by hypoxia (see above), the induction of BiP by CEMIP is likely to be an important component of the cellular response to hypoxia, allowing cancer cells to survive and metastasize [122]. 

CEMIP-induced BiP expression and activity appear to contribute to the metastasis-inducing effects of CEMIP at a number of levels. For example, CEMIP can activate the STAT3 signaling pathway in a BiP-dependent manner, thereby promoting the proliferation and migration of breast cancer cells [30,123]. In addition to inducing BiP expression, CEMIP also forms a stable complex with BiP in the ER via its B domain, which is required for CEMIP-mediated migration [29]. The formation of this complex results in Ca^2+^ leakage from the ER into the cytoplasm and the activation of protein kinase C alpha (PKCα) [29], consistent with the role of PKCα in regulating cell migration [29,124,125]. As intracellular Ca^2+^ regulates a plethora of signaling pathways [126], CEMIP-mediated Ca^2+^ leakage may thus underlie some of the metastasis-inducing effects ascribed to CEMIP expression. CEMIP also increases intracellular Ca^2+^ levels through other mechanisms. For example, in prostate cancer cells, CEMIP physically interacts with 1,4,5-trisphosphate receptor type 3 (ITPR3), a protein that regulates intracellular Ca^2+^ release [59,66]. The ensuing elevated Ca^2+^ levels lead to the increased transcription of SLC7A11, a glutamate/cysteine antiporter that is able to protect cells from ferroptosis, a form of iron-dependent programmed cell death important in matrix-detached cells [66]. 

### 3.3. EMT

EMT is an evolutionarily conserved morphogenetic program that plays numerous roles during embryogenesis and which can be reactivated in cancer cells [127]. EMT is thought to promote metastasis through a number of mechanisms, including endowing cancer cells with modified adhesive properties, enhanced motility and invasion, resistance to therapy and other pro-apoptotic stimuli, and the acquisition of stem cell characteristics [128]. The expression of CEMIP has been demonstrated to promote EMT through a number of mechanisms, as outlined below. Consistently, CEMIP is among one of eight EMT-related genes chosen for a risk assessment model that predicts the risk for metastatic breast cancer to the bone [129].

In colorectal and breast cancer, CEMIP expression inversely correlates with E-cadherin expression and is found in the invasive edge of tumors [17,29]. In gastric cancer cells, CEMIP fosters the expression of several downstream targets of Wnt/β-catenin signaling, including EMT-related molecules such as snail, vimentin, N-cadherin, c-Myc, cyclin D1, and MMPs (MMP2, MMP7, and MMP14) [22,23]. As CEMIP is also a Wnt target gene [7,59], this suggests that CEMIP serves to positively reinforce Wnt-induced EMT. CEMIP also promotes EGF-induced EMT in cervical cancer cells and in sorafenib-resistant HCC cells [41,64]. In non-small lung cancer cells, CEMIP-induced EMT has been found to be mediated via PI3K/Akt signaling [19], a downstream effector of EGFR signaling. Consistent with an EMT-inducing role, the knockdown of CEMIP in a number of settings has been reported to increase the epithelial characteristics of cells and decrease their mesenchymal properties, as evidenced by the increased expression of E-Cadherin, ZO-1, and occludin and by a decrease in vimentin, N-cadherin, and snail expression [8,25,29,41,73].

EMT may also conceivably be induced by CEMIP indirectly through its hyaluronidase activity. Notably, the small HA oligosaccharides produced through this hyaluronidase activity are able to induce the expression of MMPs [84,87,130]. The MMP-mediated release of growth factors and cytokines tethered to the extracellular matrix can induce EMT [131]. Furthermore, CEMIP also physically interacts with ephrin type-A receptor 2 (EphA2) [59], an Eph family member that can promote EMT [132] although whether this interaction regulates EMT remains to be investigated. 

### 3.4. Reprogramming of Metabolism

Metabolic reprogramming is a hallmark of cancer and involves altered glycolysis, glutaminolysis, and one-carbon and lipid metabolism [133,134]. There is increasing evidence that the dynamic adaptation of key metabolic programs is also required for the formation of metastases, by allowing disseminating tumor cells to survive their metastatic journey and successfully colonize secondary organs [135]. A number of observations suggest that CEMIP expression may contribute to the metabolic adaptation required for tumor progression and therapy resistance. For example, the positive feedback loop between β-catenin and CEMIP expression (see above) results in the upregulation of glutaminase 1 and glutamine transporters SLC1A5 and SLC38A2, which stimulates metastasis formation [38]. Furthermore, the depletion of CEMIP in prostate cancer cells reduces the expression of pyruvate dehydrogenase kinase isoform 4 (PDK4) and lactate dehydrogenase A (LDHA), enzymes important for efficient glycolysis in tumor cells, thereby decreasing cellular pyruvate, lactate, and ATP levels. This not only blocks metabolic reprogramming and impairs anoikis tolerance but also reduces migration and invasion [115]. Furthermore, through its interaction with BiP, CEMIP expression increases glucose uptake by tumor cells, which promotes their survival under hypoxic conditions [121]. Moreover, CEMIP interacts with the glycogen phosphorylase kinase β-subunit (PHKB) via its C-terminal region [65]. As a consequence, the phosphorylation of PHKB is increased, resulting in the activation of glycogen phosphorylase activity and accelerated glycogen breakdown, which promotes cancer cell survival and suppresses apoptosis [65]. In this context, it is interesting to note that the enzymatic activity of PHK is regulated by intracellular Ca^2+^ levels, which can be upregulated by CEMIP (see above) [136]. Finally, CEMIP physically interacts with and reactivates MEK1 in intestinal organoids resistant to the MEK1/2 inhibitor selumetinib, and CEMIP-dependent metabolic reprogramming has been found to contribute to the selumetinib resistance by increasing the levels of lactate and multiple amino acids [47].

## 4. Role of CEMIP in Shaping the Cancer Microenvironment

To date, virtually all efforts aimed at understanding the tumor- and metastasis-promoting roles of CEMIP have focused on its function in cancer cells. Nevertheless, a number of observations suggest that CEMIP may also contribute to tumor growth and the process of cancer dissemination through its expression in tumor-associated stromal cells, which serve to shape a pro-tumorigenic and metastasis-promoting microenvironment.

### 4.1. CEMIP-Containing Exosomes Promote Brain Metastasis

In a landmark paper, Lyden and colleagues have recently demonstrated that tumor-derived exosomal CEMIP can contribute to the formation of a pre-metastatic niche and metastasis formation in the brain [27]. In this study, breast cancer cells that form brain metastases in experimental mice produce exosomes containing CEMIP, in contrast to exosomes from related breast cancer cells that do not metastasize to the brain. Loss- and gain-of-function experiments show that exosomal CEMIP is necessary for the education of the brain microenvironment to support metastatic growth, in particularly through the reprogramming of endothelial cells and microglia. This reprogramming results in modified inositol metabolism and inflammatory cytokine expression associated with vascular co-option, invasion, and metastatic colonization in the brain microenvironment [27]. Consistently, CEMIP expression in human primary breast and lung cancers has been found to correlate with metastasis formation and poor prognosis, and brain metastases have been found to exhibit significantly higher CEMIP levels [27].

### 4.2. Putative Role for CEMIP in Regulating HA Metabolism in Cancer-Associated Fibroblasts (CAFs)

CAFs are a heterogeneous population of activated fibroblasts that often comprise the major stromal component of the tumor microenvironment. They exert a well-documented modulatory role in metastasis formation through a variety of mechanisms, including the synthesis and remodeling of the ECM, the secretion of growth factors and cytokines, and the induction of angiogenesis [137]. They also produce large quantities of HA, which stimulates both their own motility and that of tumor cells in the cancer microenvironment [138].

CEMIP is strongly expressed by a number of fibroblast subpopulations in various organs and diseases, with the highest CEMIP expression being observed in activated fibroblasts from inflammatory diseases such as Crohn’s disease, osteoarthritis, and idiopathic pulmonary fibrosis, which regulates fibroblast proliferation and migration [46,90,139]. Moreover, CEMIP plays an important role in regulating HA metabolism in these cells [3,108,140]. Notably, the CEMIP-dependent degradation of HA by fibroblasts in inflammatory diseases such as Crohn’s disease and osteoarthritis creates a pro-inflammatory environment due to the accumulation of degraded HA oligosaccharides [90,139]. Although it remains to be shown which, if any, CAF subpopulations express CEMIP, given the high levels of HA produced by CAFs, it is interesting to speculate that CEMIP expression in CAFs may promote metastasis formation through the creation of a pro-metastatic inflammatory environment via HA degradation and the accumulation of bioactive HA oligosaccharides.

### 4.3. Regulation of Tumor- and Metastasis-Promoting Inflammation by CEMIP

An inflammatory microenvironment plays a critical role in CRC development and has been associated with all stages of cancer development and malignant progression [141]. While the immune system can suppress tumor growth and metastasis formation, tumor-promoting inflammation can establish an immunosuppressive microenvironment that promotes tumorigenesis and metastasis through skewing immune cell differentiation and recruitment [142]. Examples include tumor-associated macrophages (TAMs), tumor-associated neutrophils (TANs), myeloid-derived suppressor cells (MDSCs), and regulatory T cells (T_reg_) [143]. There are strong similarities between inflammatory responses that promote metastasis and those in chronic inflammation [144]. 

CEMIP expression has been shown to regulate immune responses in a number of ways. First, it has been shown to play an important role in stimulating chronic inflammation in diseases such as arthritis, most likely through its hyaluronidase activity [139,145,146,147,148]. At the functional level, CEMIP deficiency has been shown to suppress experimentally induced arthritis in mice, which can be reversed through adenoviral-delivered CEMIP in a manner that depends on the G8 domain [60]. Neutralizing anti-CEMIP antibodies reduces levels of small HA oligosaccharides, inhibits the expression of pro-inflammatory cytokines such as TNFα, IL-1β, and IL-6, and also improves clinical and histological scores [60]. 

A number of observations support the notion that CEMIP might also contribute to a tumor- and metastasis-promoting inflammatory and immunosuppressive microenvironment. In patients with metastatic CRC, the expression levels of CEMIP correlate with neutrophil numbers in the liver. Similar observations have been made in experimental mice. Mechanistically, CEMIP expression induces chemokine production, which leads to the infiltration and polarization of immunosuppressive neutrophils to the liver and the exclusion of CD8+ T cells, thereby contributing to liver metastasis [18]. In glioblastomas, CEMIP has been shown to contribute to macrophage migration, M2 polarization, and the release of pro-tumorigenic factors that promote tumor growth. Interestingly, in this model, the loss of CEMIP expression in stromal cells reduces macrophage infiltration into the glioblastoma tissue, which is linked to an accumulation of HA [28]. Notably, the degraded HA oligosaccharides produced by CEMIP can induce the M2 polarization of macrophages [85]. Furthermore, small HA oligosaccharides can act as DAMPs (see above); their sustained production, therefore, has the potential to induce chronic inflammation, which can act to potentiate metastasis formation. 

The ability of CEMIP to induce EMT (see above) may also be relevant in this context, as EMT stimulates the production of pro-inflammatory factors by cancer cells [149]. Conversely, chronic inflammatory microenvironments stimulate EMT in the context of cancer [150]; thus, if CEMIP expression contributes to creating such microenvironments, this may be another means through which it promotes EMT.

## 5. Therapeutic Implications

CEMIP plays a role in the pathogenesis of several diseases, including cancer. Accordingly, a number of approaches have been and are being developed to inhibit its expression and activity. Only a few studies have investigated the utility of inhibiting CEMIP to combat cancer and metastatic disease, and given the functional role that CEMIP plays in the process of tumorigenesis and metastasis, this could be a promising therapeutic avenue to explore in the future.

A number of approaches have been taken to inhibit the CEMIP hyaluronidase activity. Antibodies that neutralize the hyaluronidase activity of CEMIP have shown utility in the context of experimentally induced arthritis [60], and it would be interesting to see whether a similar approach might have an impact in the context of tumorigenesis and metastasis. Our own work indicates that sulfated HA derivatives are potent inhibitors of the CEMIP hyaluronidase activity; they may, therefore, also have clinical utility [151]. 

Other approaches have focused on the inhibition of CEMIP expression. Although it is unlikely that these approaches are specific to CEMIP, they have shown promising therapeutic results. In triple-negative breast cancer, the loss of H3K27me3 at the CEMIP promoter via H3K27me3 demethylase results in upregulated CEMIP expression. Consistently, treatment with GSK-J4, an inhibitor of the demethylase, downregulates CEMIP expression and concomitantly decreases tumor growth and migration [152]. Cyclooxygenase-2 (COX-2) inhibitor NS398 suppresses CEMIP expression in colorectal adenomas [153], decreases CEMIP expression, and hinders migration in pancreatic cancer cells [91]. The therapeutic delivery of CEMIP-regulating miRNAs may also have utility, as the enforced expression of miR-216a downregulates CEMIP in colorectal cancer cells and as a result suppresses tumor invasion and metastasis [8]. The multi-kinase inhibitor lenvatinib has also been reported to suppress CEMIP expression in gallbladder cancer through targeting AKT signaling [154]. Recently, the expression of CEMIP has been more specifically targeted using the hydrogel-based delivery of CEMIP-specific short hairpin RNAs, which elicited anti-tumor and –metastasis effects in a spontaneous breast tumor model [155].

Pirfenidone is an anti-fibrotic and anti-inflammatory agent used to treat idiopathic pulmonary fibrosis, which suppresses CEMIP levels and increases HA levels in the lungs of patients [46]. Recent data suggest that pirfenidone might also be useful for treating cancer. In MC38 colon adenocarcinoma cells, pirfenidone suppresses CEMIP expression and improves the response to PD-1 blockade in experimental animals through fostering the infiltration of CD8+ T cells, which is regulated at least in part by CEMIP [18]. In addition to having an impact on immune cells, pirfenidone is also able to inhibit fibroblast activation, thereby influencing the tumor–stroma crosstalk and suppressing tumor progression [156].

A major problem in the clinical management of cancer is the development of resistance to chemotherapy and targeted therapies. Resistance to therapy allows metastases to form, leading to the subsequent demise of the patient. CEMIP has been implicated in contributing to therapy resistance. For example, increased CEMIP expression is functionally associated with resistance to the multi-kinase inhibitor sorafenib in hepatocellular cells, which results in increased invasion and metastasis [41]. Accordingly, a number of observations suggest that targeting CEMIP expression and activity can re-sensitize tumors to therapeutic intervention. In the context of chemotherapy, the downregulation of CEMIP increases the sensitivity of resistant gastric cancer cells to the chemotherapeutic agent 5-fluorouracil (5-FU), indicating the potential utility of using CEMIP inhibitors in combination with chemotherapy [25,72]. Resistance to tyrosine kinase inhibitors that target the MET oncogene ensues when neuropilin-2 is downregulated. As a consequence, CEMIP expression is increased and in turn enforces EGFR expression that mediates therapy resistance [119]. The silencing of CEMIP in resistant tumor cells improves the therapeutic response [119]. 

Cancers such as colorectal carcinomas and melanoma can be driven by BRAFv600E mutations, making them sensitive to BRAF inhibitors such as vemurafenib. However, innate or acquired resistance to this therapy limits their effectiveness, facilitating the outgrowth of metastases. The additional use of MEK inhibitors can help to combat resistance to BRAFi, although again, innate and acquired resistance to MEKi ultimately limits the efficacy of this approach. Increased levels of endosomal CEMIP as a consequence of Wnt signaling have been implicated in mediating resistance to MEK inhibitors in BRAF^V600E^-mutated colorectal cancers, which is mediated by the binding of CEMIP to MEK1 and subsequent sustained ERK1/2 activation [47]. Thus, targeting CEMIP in this context also holds promise for overcoming therapy resistance.

## 6. Conclusions/Perspectives

Since its discovery as a theoretical protein with unknown function, CEMIP has emerged as a multifaceted player in a wide-range of physiological and pathological processes. In the context of cancer, and most notably for CRC, CEMIP has emerged as a strong candidate diagnostic marker, and has remarkable potential as an indicator of prognosis. While the literature underpinning this conclusion is compelling, considerable additional work will be required to develop CEMIP into a biomarker that can be used routinely in the clinic. Given that CEMIP nucleic acid and protein can be detected in the blood of CRC patients, the development of reliable and consistent blood-based assays with good sensitivity and specificity would offer the most likely route to successful clinical application. Studies with larger patient cohorts are needed to help to define for which disease stages and subtypes CEMIP detection is diagnostically/prognostically relevant. This would determine whether monitoring CEMIP in the blood has potential, for example, in screening healthy populations to identify individuals with occult disease, in staging/stratification purposes to improve patient outcome by guiding therapeutic interventions, and/or in evaluating the efficacy of therapeutic intervention and subsequent remission longitudinally. 

The wealth of literature summarized in this review attests to the fact that CEMIP is a functional biomarker that plays an important role in tumor progression and metastasis. This clearly identifies CEMIP as a promising therapeutic target. Nevertheless, a detailed understanding of the molecular activities of CEMIP, their role in cancer biology, and the domains of the protein that contribute to these activities is currently lacking. If the potential of CEMIP as a therapeutic target is to be capitalized on, a careful and thorough dissection of CEMIP molecular biology in the context of cancer is a pre-requisite for the appropriate targeting of the protein. Although the derivation of the name CEMIP suggests that the major role of the CEMIP protein is in regulating hyaluronan metabolism and cell migration, it is now clear that CEMIP does much more than this, in particular in the context of signal transduction and the regulation of gene expression. New functional roles continue to be discovered for CEMIP. Furthermore, most studies to date have investigated the tumor- and metastasis-promoting role of CEMIP from the perspective of CEMIP expression in cancer cells. However, as pointed out above, CEMIP may equally contribute to tumorigenesis and metastasis formation through expression in cells in the tumor microenvironment. This possibility is in an early stage of investigation, and considerable further in-depth studies are required.

Two pivotal open questions concern the hyaluronidase activity ascribed to CEMIP, which have an impact not only on a mechanistic understanding of how the CEMIP protein contributes to metastasis formation but also on efforts to target the protein and its activity therapeutically. First, to what degree are the metastasis-promoting roles of CEMIP dependent on its hyaluronidase activity? Second, and more fundamentally, does the CEMIP protein itself have the capacity to directly depolymerize HA? In this latter regard, although CEMIP can bind to HA, it is notable that the CEMIP protein does not possess any known hyaluronidase domains. Furthermore, the purified CEMIP protein is unable to depolymerize HA and requires the presence of cells or cell membranes to degrade HA. It is possible that an as-yet-unrecognized domain in CEMIP is able to depolymerize HA, and that this activity needs to be “switched on” through interacting with other cellular components. However, given that CEMIP interacts with a wide range of proteins (Table 1), it is also conceivable that CEMIP might exert hyaluronidase activity through interacting with and regulating the activity of another as-yet-unidentified protein that then exerts the HA depolymerizing function. To differentiate among these possibilities, the molecular biology of CEMIP hyaluronidase activity needs to be carefully dissected, for example, through mutational analyses, the development of specific inhibitors, and the characterization of the molecular partners required for CEMIP to elicit HA polymerization.

The effective therapeutic targeting of CEMIP also requires further work to understand the molecular function of the CEMIP protein in the context of the many proteins that it is known to interact with. The domains of CEMIP required for these interactions, as well as their regulatory function and relevance at the molecular level, need to be defined. In conjunction with additional studies to determine which of the molecular partners of CEMIP are decisive for CEMIP-mediated tumor progression, this would in turn facilitate a deeper understanding of how CEMIP contributes to the process of tumor growth and metastasis formation and would identify novel therapeutics targets that can be leveraged to combat CEMIP-dependent metastasis and therapy resistance. 

## Figures and Tables

**Figure 1 cancers-14-05093-f001:**
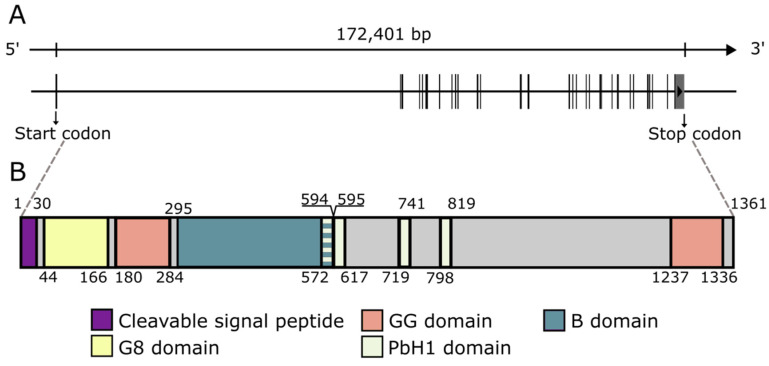
**The genomic organization and protein domains of CEMIP.** (**A**). The genomic organization of the 29 exons that comprise the CEMIP gene. (**B**). The major domains that comprise the CEMIP protein. The numbers refer to the amino acid positions that flank the different domains. The length and position of exons as well as domains are true to scale.

**Figure 2 cancers-14-05093-f002:**
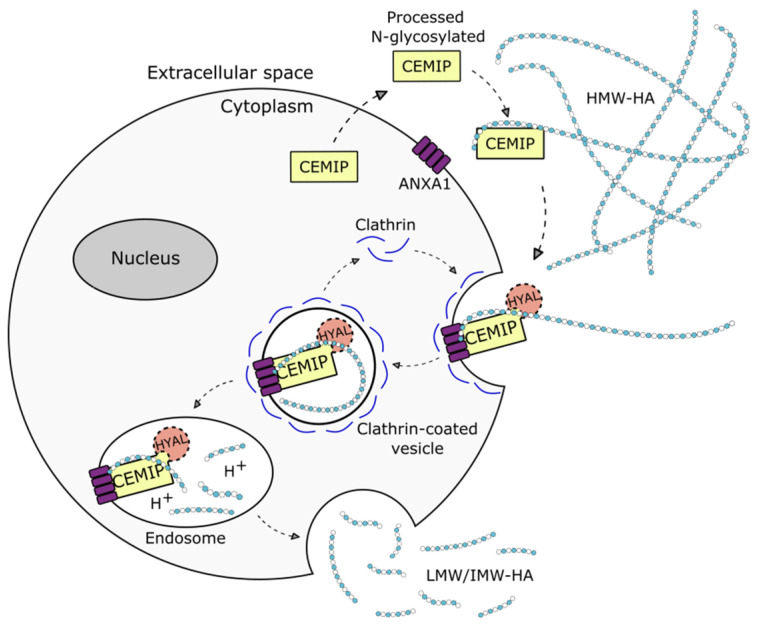
**Schematic representation of the proposed mechanism of action for CEMIP-mediated degradation of hyaluronan (HA).** Extracellular CEMIP binds to high molecular weight (HMW) HA and internalizes it via clathrin-coated endocytosis after binding to annexin A1 (ANXA1). In the endosome, HMW-HA is degraded into low and intermediate (LMW/IMW) HA. The degraded HA fragments are subsequently released back into the extracellular milieu. HYAL represents a hypothetical bone fide hyaluronidase that might conceivably participate in CEMIP-mediated HA degradation.

**Figure 3 cancers-14-05093-f003:**
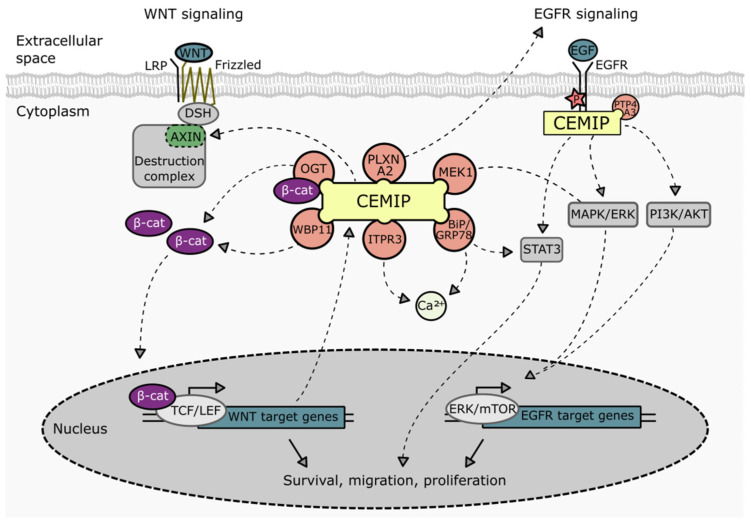
**Schematic representation of the involvement of CEMIP in various signaling pathways.** Direct interaction partners of CEMIP related to Wnt, EGFR, and Ca^2+^ signaling pathways are indicated. BiP: binding immunoglobulin protein. ITPR3: 1,4,5-trisphosphate receptor type 3. MEK1: Mitogen-activated protein kinase kinase 1. OGT: O-GlcNAc transferase. PLXNA2: Plexin A2. PTP4A3: protein tyrosine phosphatase 4A3. WBP11: WW domain binding protein 11.

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
