# Peer review of "CEMIP, a Promising Biomarker That Promotes the Progression and Metastasis of Colorectal and Other Types of Cancer"

_cancers, 2022, doi:10.3390/cancers14205093_

Round 1

Reviewer 1 Report

The authors of this review present a well laid out, thoughtful, and thorough summary of the literature and findings pertaining to the role of CEMIP in cancer progression and metastasis, and both its potential as a biomarker in colorectal and other cancers as well as a therapeutic target. The authors break the review into logical sections that help facilitate the reader in understanding the complexity of findings for CEMIP (for example, the numerous potential interacting partners and putative pathways of involvement). The review touches on the appropriate questions that need addressing in CEMIP research (particularly on how CEMIP is facilitating hyaluronan degradation and what role does this contribute to cancer progression and metastasis) and provides an especially nice section covering therapeutic implications of targeting CEMIP. To date, CEMIP as a potential therapeutic target has not been well addressed in the literature. In fact, there has been little, review-wise, that has been well-written on CEMIP’s role in cancer or its therapeutic and biomarker implications. Finally, the authors have done an excellent job of capturing the necessary and appropriate references covering what is known about CEMIP’s biological functions, its potential utility as a diagnostic and prognostic biomarker, and implications as a therapeutic target.

There is one minor suggestion that would further strengthen the already excellent review. The title mentions CEMIP as a promising biomarker, yet the authors discuss this aspect in only two paragraphs in the introduction. CEMIP expression levels are highly induced in many epithelial cancers at the early stages of disease, making it a very strong candidate as an early diagnostic biomarker. Likewise, the literature on CEMIP as a potential prognostic marker is compelling. A brief paragraph in the summary touching on the importance of continued studies into the potential of CEMIPs utility in these biomarker areas would be a nice addition. If CEMIP RNA, protein, or cancer-specific protein modification could be utilized as an early diagnostic, the impact on cancer prevention would be significant. Likewise, if confirmed as a prognostic biomarker, it could significantly impact treatment decisions in a variety of cancers. Philosophically, one could almost argue that CEMIP might have greater importance as a diagnostic and prognostic biomarker than as a therapeutic target.

Author Response

Thank you very much indeed for these very positive comments about our review article, and for the excellent suggestion regarding inclusion of an additional paragraph in the Conclusion/perspective section about the potential of CEMIP as a biomarker. In the revised version of the manuscript we have included the paragraph as suggested.

Reviewer 2 Report

This review summarizes the effects of cell migration–inducing and hyaluronan-binding protein (CEMIP) on promoting the process of tumor growth and metastasis. The manuscript is well constructed with abundant relevant examples, and the examples are well explained with great clarity. However, the manuscript could benefit from a careful check by an expert proofreader, because some small grammatical and style changes are required to make it more readable. Overall, this review could be of value for a better understanding of how the CEMIP contribute to tumorigenesis and metastasis formation, but some more details must be addressed before publication.

I have the following comments for this manuscript:

1. In the Abstract section, please correct the sentence “CEMIP is a promising biomarker for colorectal cancer, is associated with poor prognosis for patients suffering from colorectal and other types of cancer, and functionally contributes to tumor progression and metastasis”, as the second sentence lacks subject.

2. Please add a table which summarizes the applications of CEMIP inhibitors for cancer treatment, including inhibitors, cancer types, outcomes, advantages, and disadvantages.

3. Please move the table 1 into the main text.

4. At Line 183-190, please indicate clearly how many times dose the CEMIP expression increase in the cancer tissues compared with normal organs, and detailed differences (values) between various cancers.

5. Please change the title of 3.2.1-3.2.3 to make them more closely related to the previous paragraph. For example, use Wnt signaling pathway, EGFR signaling pathway, and Ca2+ signaling pathway.

6. Please change the title of 4, 4.1-4.3 to give a summary for the role of CEMIP on changing the cancer microenvironment.

7.Please rewrite the Conclusion/perspectives section to make it more focused on the topic, including the role of CEMIP on promoting tumor progression and metastasis, future prospectives in the development of novel therapeutics.

Author Response

Thank you very much for these positive and constructive comments. The second author is a native speaker and educated at the University of Cambridge in the UK. He has now carefully proofread the manuscript again and removed the minor grammatical errors mentioned by the reviewer. Notably, the original text contained a hybrid of both British and American English spelling conventions. The revised text has been edited such that only American English spellings are used. Below we outline our comments to points 1 – 7 raised by the reviewer.

  1. The sentence highlighted by the reviewer lists three features of CEMIP expression in the context of cancer, is grammatically correct and does not need correcting. Nevertheless, in view of the reviewer’s comments we have now divided the sentence in question into two sentences, with the aim of improving the readability of the text.
  2. As CEMIP inhibitors have only recently been described in experimental settings, there is currently no clinical application of CEMIP inhibitors in the context of cancer treatment. Therefore the requested table cannot be provided.

  1. Table 1 has been moved into the main text as requested.

  1. Representative values have been added.

  1. The titles have been modified.

  1. The titles have been modified.

  1. The Conclusions/perspectives section has been augmented to take into account the reviewer’s comments.

We hope that the changes made to the manuscript mean that it can now be accepted for publication.

Round 2

Reviewer 2 Report

The authors have satisfactorily responded to all my comments and made necessary changes to the manuscript. As technical points have been clarified, no major points left.